# CLOOME: contrastive learning unlocks bioimaging databases for queries with chemical structures

Ana Sanchez-Fernandez[1], Elisabeth Rumetshofer[1], Sepp Hochreiter [1,2] & Günter Klambauer [1] ✉

The field of bioimage analysis is currently impacted by a profound transformation, driven by the advancements in imaging technologies and artificial intelligence. The emergence of multi-modal AI systems could allow extracting and utilizing knowledge from bioimaging databases based on information from other data modalities. We leverage the multi-modal contrastive learning paradigm, which enables the embedding of both bioimages and chemical structures into a unified space by means of bioimage and molecular structure encoders. This common embedding space unlocks the possibility of querying bioimaging databases with chemical structures that induce different phenotypic effects. Concretely, in this work we show that a retrieval system based on multi-modal contrastive learning is capable of identifying the correct bioimage corresponding to a given chemical structure from a database of ~2000 candidate images with a top-1 accuracy >70 times higher than a random baseline. Additionally, the bioimage encoder demonstrates remarkable transferability to various further prediction tasks within the domain of drug discovery, such as activity prediction, molecule classification, and mechanism of action identification. Thus, our approach not only addresses the current limitations of bioimaging databases but also paves the way towards foundation models for microscopy images.

Biological and chemical databases and their querying mechanisms are at the heart of research in molecular biology. Sequence databases, such as RefSeq[1] or UniProt[2], contain DNA or protein sequences, and are often queried with a given sequence using BLAST[3] or its variants. Genome databases[4] usually allow for multiple types of querying mechanisms, such as genetic location, gene names, or accession numbers. Protein structure databases, for example, the Protein Data Bank (PDB)[5], offer a range of querying mechanisms from sequence similarities to structural queries based on 3D shape. The chemical databases ChEMBL[6] and PubChem[7] are huge corpora of chemical structures that contain billions of small molecules. The International Chemical Identifier (InChI)[8] was designed to facilitate searching for chemical structures in such

databases, which is difficult because of the graph matching problem. While BLAST, the structural search in PDB, and the InChI-based queries can be considered as associative or content-based querying, bioimaging databases still rely on manual annotation and text-based search. However, querying large bioimaging databases with a chemical structure that induces the phenotypic effect captured by the image could considerably empower biomedical research. Additionally, unlocking chemical databases for queries with a microscopy image capturing the phenotypic effects of a chemical structure could be equally important. (see Fig. 1a, b).

Recently, contrastive learning has emerged as a powerful paradigm to learn rich representations from data[9]. The contrastive learning

[1]ELLIS Unit Linz and LIT AI Lab, Institute for Machine Learning, Johannes Kepler University Linz, Linz, Austria. [2]Institute of Advanced Research in Artificial Intelligence (IARAI), Vienna, Austria. ✉e-mail: klambauer@ml.jku.at

Fig. 1 | **Overview of the CLOOME framework. a**, **b** The CLOOME encoders can be used to query a bioimaging database (**a**) by a chemical structure, and vice versa, query a chemical database by a microscopy image (**b**). **c** Visualization of the embedding space in terms of a t-SNE projection of image embeddings of new cell phenotypes. Each point represents a microscopy image from a hold-out set. The color indicates the cell phenotype, which was also withheld from training. The CLOOME embeddings (left) are indicative of the cell phenotype (clustered colors). CellProfiler features are less indicative of cell phenotypes (only a few colors cluster together). **d** A multi-modal setting for imaging cell phenotypes. Small molecules are administered to cells which are then imaged to capture potential phenotypic effects. In this way, matched image-structure pairs are obtained. **e** Schematic depiction of the training procedure of CLOOME. During training, the similarity of matched image-structure pairs is increased (black arrows), while the similarity of un-matched image-structure pairs is decreased (gray arrows). **f** The encoders of CLOOME map chemical structures and microscopy images to the same embedding space using a structure and a microscopy image encoder. Both encoders are deep neural networks. Matched pairs of chemical structures and microscopy images are mapped to embeddings that are close together, whereas un-matched pairs are mapped to embeddings that are separated. Source data are provided as a Source data file.

methods CLIP and CLOOB embed natural language and images into the same representation space[10,11]. Contrastive learning enforces that images and their matched text captions are close to one another in this embedding space, while un-matched images and captions are separated. Therefore, text prompts can query an image database by extracting nearby images in the embedding space and vice versa[10]. These text-image embedding spaces enabled the generation of realistic images from short text prompts and led to the recent emergence of "AI art"[12]. In this work, we use these powerful contrastive learning paradigms to enable querying or retrieval systems for bioimaging databases.

Microscopy imaging has been used as an informative and time- and cost-efficient biotechnology to characterize cell phenotypes, tissues, or cellular processes[13,14]. Consequently, there have been substantial efforts by the scientific community to use high-throughput microscopy imaging[15] as informative read-out and characterization of cellular systems and cell phenotypes under diverse perturbations[14,16]. For instance, features computed from microscopy images[17] have already been proved useful to find sets of compounds with higher diversity performance in high throughput screening assays than those from chemical structures[18]. In addition to the wealth of information that is comprehensible and informative for human experts, these

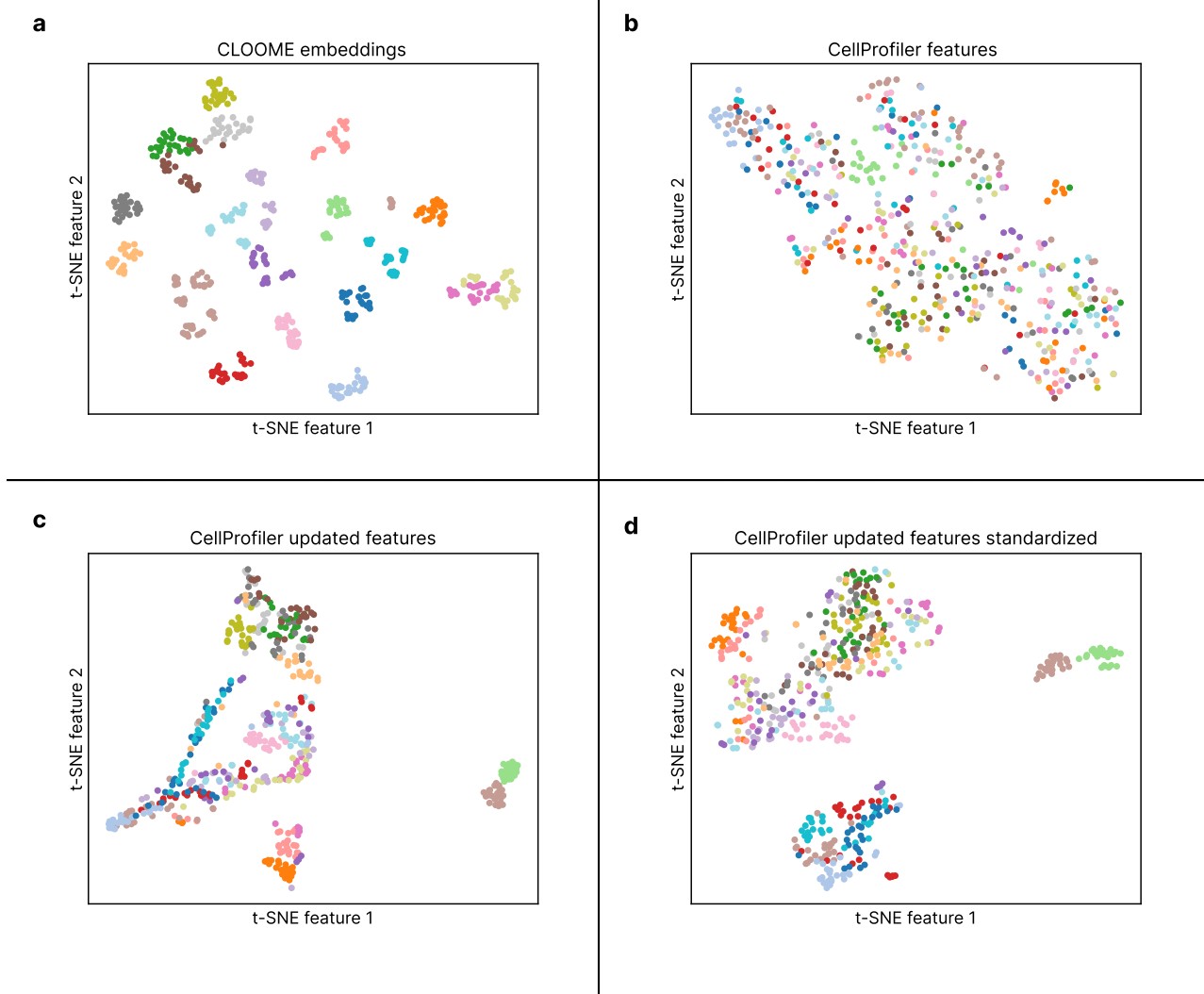

**Fig. 2 | Two-dimensional downprojection of CLOOME embeddings and CellProfiler features.** t-SNE downprojection of the CLOOME learned embeddings (**a**), the CellProfiler extracted features provided in ref. 24 (**b**), and the CellProfiler extracted features using an updated pipeline, non-standardized and standardized (**c** and **d**, respectively) of all microscopy images corresponding to 20 randomly selected molecules from the test set. The colors represent different molecules. Source data are provided as a Source data file.

microscopy images also contain large amounts of biological information inaccessible to humans, but which can be successfully extracted by computational methods, such as Deep Learning[19]. The immense amount of microscopy images are stored in large bioimaging databases, many of which are publicly available. Their querying mechanisms, however, are still limited to queries by textual annotations. A common embedding space of (a) microscopy images capturing phenotypic effects of perturbations, and (b) chemical structures inducing those effects would allow for associative or content-based querying of both imaging and chemical databases. Such an embedding space would represent cellular processes both in terms of the chemical structures that induce them and in terms of images that capture the cell phenotypes caused by these processes. New applications such as the detection of novel cell phenotypes are possible through such embedding spaces (see Fig. 2a and Fig. 3).

CLIP and CLOOB models have been constructed via contrastive learning on large image-text datasets[10]. Analogously to these image-text datasets, the Cell Painting dataset[16] contains image-structure pairs (see Fig. 1d). Therefore, we were able to use contrastive learning to jointly train a microscopy image encoder and a chemical structure encoder to construct a common embedding space of microscopy

images capturing cell phenotypes and chemical structures representing the perturbations. We propose a contrastive learning framework for image-structure pairs that we call CLOOME (see Fig. 1d, e, f). Within the CLOOME framework, a microscopy image encoder and a chemical structure encoder are learned by contrasting representations of matched image-structure pairs against un-matched examples from other pairs. Because our framework extends the contrastive learning methods CLIP[10] and CLOOB[11] to image-structure pairs, we call it Contrastive Learning and leave-One-Out-boost for Molecule Encoders (CLOOME).

We introduce CLOOME as a novel tool that enables querying large chemical databases with microscopy images, and vice versa. Moreover, the experiments conducted in this study show that our method is useful beyond this cross-modal querying, because the encoders provide rich and transferable embeddings of bioimages. In many cases these embeddings are more informative than high content imaging platforms[17], for example, for bioactivity prediction and mechanism of action prediction (see the "Results" section). In summary, we illustrate four use case scenarios that we envision for CLOOME.

1. A query molecule is known to cause an interesting biological effect, and the goal is to find other chemical structures that show the same biological activity. If a microscopy image of this query molecule

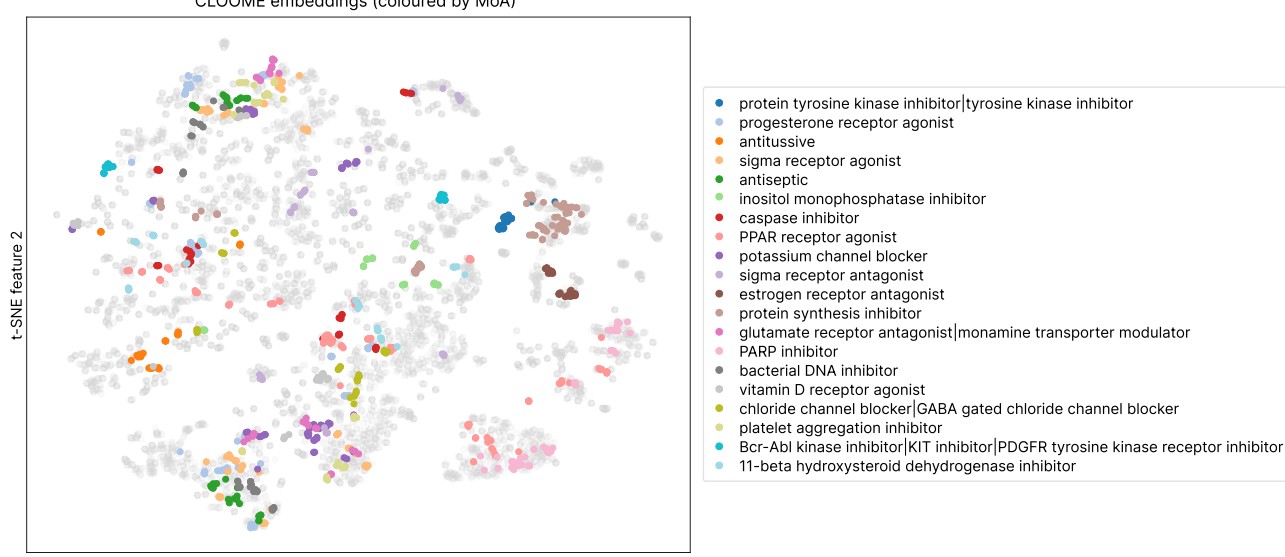

**Fig. 3 | t-SNE downprojection of the learned CLOOME image embeddings.**
Samples corresponding to 20 randomly selected mechanisms of action (MoA) are shown in different colors, while the remaining samples are shown in gray. Embeddings of some images capturing the same MoA are close together in the t-SNE plot, which indicates that the CLOOME embeddings could be useful for identifying new MoAs. Source data are provided as a Source data file.

is available, CLOOME could be used to retrieve chemical structures from large chemical databases. These chemical structures are candidates to exhibit the same biological activity and to show an equivalent phenotype as the one shown in the query molecules. Note that this type of retrieval does not need any text or annotation and, to our knowledge, has not been suggested in previous literature (see structure retrieval results in the section "A retrieval system for imaging and chemical databases").

Moreover, this application could also be of interest for biological effects, for which no molecules are currently known. Assume that the biological effect concerns the inhibition of a certain protein. A gene knock-out procedure could be carried out for this protein (e.g., using CRISPR-Cas9) and the resulting modified cells, captured with microscopy. Using this microscopy image as a query, CLOOME could be used to search for chemical structures that induce the inhibition of this specific target. In contrast to target-based virtual screening, this procedure involves the full complexity of a biological system.

2. For a new chemical structure it is unknown which cellular process it induces. The molecule's chemical structure is used to query a database of microscopy images of cells treated with different chemical compounds that have well-known biological effects. CLOOME could be used to retrieve images showing the potential phenotype of the query molecule, which could give an indication about the induced cellular process. This type of querying mechanism has also not been suggested before (see image retrieval results in the section "A retrieval system for imaging and chemical databases").

3. CLOOME could also be used to predict the activity of a compound in a given assay. Even though this task has been previously addressed by classical supervised learning methods[19,20], the training process was compute-intensive and slow. CLOOME's pre-training reduces this task to fitting a logistic regression model, which is compute-efficient and fast. After making predictions with CLOOME for a set of molecules, the results could be used to guide the initial screening campaigns in the drug discovery process (see the section "Bio-activity prediction as downstream task").

4. As in the second case scenario, we assume that the goal is to find the cellular process that is induced by a certain molecule. However, we use image-image relations instead of structure-image relations. We use the image treated with the molecule of interest to query an image database whose images capture well-studied mechanisms of action (MoA). Then, CLOOME retrieves the most similar images from that bioimaging database, and those are labeled by MoAs, such that the MoA of the molecule of interest can be inferred (see the section "Zero-shot image-to-image mechanism of action (MoA) classification").

In the following sections, we:
(a)    present a new contrastive learning approach for learning rich and expressive representations of microscopy images and chemical structures without the need for human annotation or bioactivity data;
(b)    we demonstrate that our framework yields a retrieval system which allows to query bioimaging databases by chemical structures linked by the underlying captured or induced cellular process, and vice versa; and
(c)    show that the learned representations are highly transferable to several relevant downstream tasks in drug discovery, such as activity prediction, microscopy image classification, and mechanism of action identification.

## Results

First, we demonstrate the abilities of CLOOME as a retrieval system for bioimaging and chemical databases (see Fig. 4a). Then, we use CLOOME's image embeddings to predict assay bioactivities by fitting merely a logistic regression model (see Fig. 4b). Lastly, we evaluate the performance of CLOOME image embeddings in molecule and mechanism of action classification tasks. Figure 4c and d depict the experimental setting for these tasks.

### A retrieval system for imaging and chemical databases

In this experiment, we assessed the ability of CLOOME to correctly retrieve the matched chemical structure given a microscopy image of cells treated with this molecule. Notably, this is an exceptionally challenging task for human experts, even considered close-to-impossible: given a microscopy image of cells, the task is to select the chemical structure with which they have been treated from a set of thousands of candidate structures. Since cells often do not exhibit any or only subtle phenotypic changes, this task is highly ambitious.

This image-based retrieval task can also be understood as a bioisosteric replacement task[21]: bioisosteres are molecules with

**a** Molecule retrieval for bioactivity matching

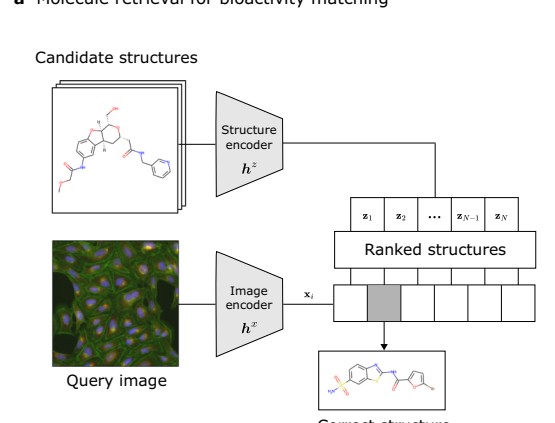

**b** Linear probing for bioactivity prediction

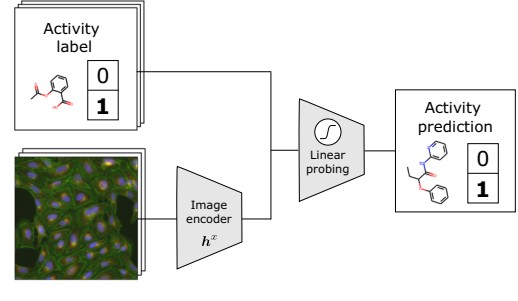

**c** Zero-shot image-to-image molecule classification

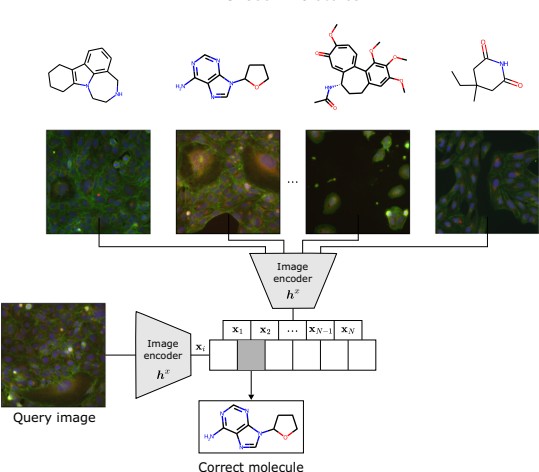

**d** Zero-shot image-to-image MoA classification

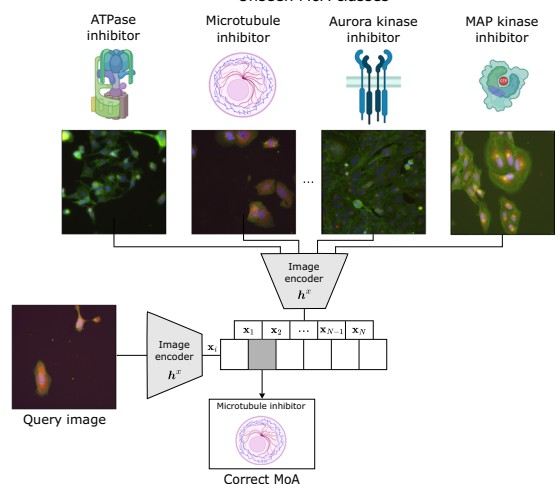

**Fig. 4 | Overview of the four different use-cases of CLOOME evaluated in this study.** An adaptive *image encoder* $h^x(.)$ and an adaptive *structure-encoder* $h^z(.)$ map the microscopy images and chemical structures to their embeddings $x_n = h^x(x_n)$ and $z_n = h^z(z_n)$, respectively. **a** Multi-modal retrieval task using CLOOME image and molecule embeddings. The resulting embeddings can be used to rank chemical structures that induce similar phenotypic effects, and vice versa. **b** Using the CLOOME embeddings for activity prediction. A logistic regression model is trained for activity prediction tasks. **c** Zero-shot image-to-image classification task using CLOOME image embeddings for molecule prediction. A set of representative images, one for each molecule, are used to infer which compound was applied in a query image. **d** Zero-shot image-to-image classification task using CLOOME image embeddings for mechanism of action (MoA) prediction. In this case, the set of representative images depict each MoA. A query image is classified into the most likely MoA category based on its similarity with a corresponding representative image. Icons representing different MoAs were created with BioRender.com.

roughly the same biological properties or activities. Bioisoterism is highly relevant in drug discovery when a molecule should be replaced with another, but at the same time its biological activity should be kept. An extreme case of bioisosteric replacement is scaffold hopping, because molecules with highly different chemical structures but still the same bioactivity should be found. With this experiment, we evaluate the ability of CLOOME to correctly rank the matched molecular structure given the corresponding image. Other high-ranked structures could be potential bioisosteres, which makes this experiment a proxy for the bioisteric replacement problems (see Fig. 4a).

As a baseline, we included a bilinear model defined by $y = x^T W z$, where $x$ and $z$ are one image and molecule feature vectors, respectively, and $W$ is a trainable weight matrix. Analogously to the CLIP method, the InfoNCE objective was symmetrically computed between modalities. The explored hyperparameters for this model can be found in Supplementary Table 6. Additionally, a random baseline is included

to demonstrate the performance improvement of the different methods in comparison to selecting the correct image or molecule by chance.

On hold-out data of 2115 image and molecule pairs, CLOOME ranked the matched molecule in the first place for 3% of the cases. A random method would achieve a value of $1/2115 \approx 0.047\%$, which indicates a ~70-fold improvement of CLOOME. For this task, different hyperparameters and model were selected based on the appropriate validation metric (see Supplementary Section 1.2). The top-1, top-5, top-10 accuracy are given in Table 1 for retrieving from a database of 2115 instances in the random split, and 1398 in the scaffold split. Additionally, we report the same metrics for a sampling rate of 1%, or equivalently, 1 matched example together with 99 un-matched ones—a setting often used to evaluate retrieval systems, see Supplementary Table 7. Further, some examples are displayed in Fig. 5. This is, to our knowledge, the first system of cell-image-based retrieval of molecular structures.

**Table 1 | Results for the retrieval task among 2115 and 1398 candidates, in the random and scaffold split, respectively**

| Split | Method | Top-k accuracy (%) | | | | | |
|---|---|---|---|---|---|---|---|
| | | Top-1 | 95%-CI | Top-5 | 95%-CI | Top-10 | 95%-CI |
| Random | CLOOME (structure retr.) | **3.78** | [3.01, 4.69] | **7.94** | [6.83, 9.18] | **9.46** | [8.24, 10.8] |
| | CLOOME (image retr.) | **3.22** | [2.51, 4.06] | **8.42** | [7.27, 9.68] | **9.88** | [8.64, 11.2] |
| | Bilinear model (structure retr.) | 0.473 | [0.227, 0.868] | 2.27 | [1.68, 2.10] | 3.92 | [3.14, 4.84] |
| | Bilinear model (image retr.) | 0.804 | [0.469, 1.28] | 3.64 | [2.51, 4.06] | 5.53 | [3.95, 5.82] |
| | Random | 0.0473 | [0.0012, 0.263] | 0.236 | [0.0768, 0.551] | 0.473 | [0.227, 0.868] |
| Scaffold | CLOOME (structure retr.) | **2.79** | [1.99, 3.79] | **6.29** | [5.08, 7.70] | **7.58** | [6.25, 9.10] |
| | CLOOME (image retr.) | **2.50** | [1.75, 3.46] | **6.58** | [5.34, 8.01] | **8.08** | [6.71, 9.64] |
| | Bilinear model (structure retr.) | 0.787 | [0.393, 1.40] | 2.72 | [1.93, 3.71] | 4.36 | [3.35, 5.57] |
| | Bilinear model (image retr.) | 0.930 | [0.496, 1.58] | 2.65 | [1.87, 3.63] | 4.51 | [3.48, 5.73] |
| | Random | 0.0715 | [0.00181, 0.398] | 0.357 | [0.116, 0.833] | 0.715 | [0.344, 1.31] |

Given a molecule-perturbed microscopy image, the matched molecule must be selected from a set of candidates, and vice versa. Top-1, top-5 and top-10 accuracy in percentage are shown for a hold-out test set, along with the upper and lower limits for a 95% confidence interval (CI) ($n$ = 2115 for the random split and $n$ = 1398 for the scaffold split) on the resulting proportion. The best method in each category is marked in bold.

## Bio-activity prediction as downstream task

In this experiment, we tested whether the representations learned by CLOOME are transferable by linear probing on 209 downstream activity prediction tasks. The *linear probing* test[22,23] on downstream tasks is often performed for contrastive learning approaches to check the transferability of learned features. In such experiments, the representations of the pre-trained encoders are used, and only a single-layer network, such as logistic regression, is fit to the given labels for the supervised task. If the linear probing test yields good predictive quality, usually below a fully supervised approach[23], the representations are considered transferable.

The prediction tasks that we employed for linear probing evaluation is the same as used in Hofmarcher et al.[20]. It is a subset of the Cell Painting dataset, consisting of 284,035 images for which the activity labels of the compound treatments were retrieved from ChEMBL. The retrieved labels correspond to 10,574 compounds across 209 activity prediction tasks, which are binary classification problems. However, activity data points are not available for all compounds in all of the tasks, which results in a sparse label matrix. The data was split into 70% training, 10% validation, and 20% test sets. This split had been carried out by grouping views from samples treated with the same molecule.

We use image features taken from the penultimate layer of the image encoder, omitting the classification layer. We train a logistic regression classifier, and report the corresponding metric for each task. The L2 regularization strength $\lambda$ was tuned individually for each one of the tasks, considering the values $\{10^{-6}, 10^{-5}, ..., 10^{6}\}$.

In order to evaluate model performance for this downstream task, we use the area under the ROC curve (AUC), which is one of the most prevalent metrics for drug discovery[19,20], as it considers the order of the molecules regarding their activity. We also show the number of tasks for which this metric is higher than the thresholds 0.9, 0.8, and 0.7, respectively. These thresholds have been used in previous studies[19,20] because models within those categories lead to certain levels of enrichment of hit rates in drug discovery projects.

As baselines, we consider methods reported in Hofmarcher et al.[20]. They are the best-performing methods for bioactivity prediction using microscopy images to date and consist of different convolutional neural network architectures, used in a fully supervised setting, and a method ("FNN") that uses expert-designed cell features[17,19,24]. The compared methods were trained in a multi-task setting to predict activity labels for 209 tasks, extracted from ChEMBL.

The predictive performance on the downstream activity prediction tasks is reported in Table 2. CLOOME reached an average AUC of $0.714 \pm 0.20$ across prediction tasks, which indicates that the learned

representations are indeed transferable since no activity data had been used to train the CLOOME encoders. CLOOME even outperformed fully supervised methods, such as M-CNN[25] and SC-CNN[20], with respect to AUC.

## Zero-shot image-to-image molecule classification

The goal of this analysis is to evaluate the potential of image embeddings of CLOOME to distinguish the specific applied molecule. Note that, instead of using both molecule and image embeddings, as in the retrieval task, only image embeddings were used in this case. However, this task is not intended to be considered a real use-case scenario, but more of a proxy task for technical validation. This evaluation can also be regarded as a quantitative evaluation of how closely clustered are image embeddings of cells treated with the same molecule.

A zero-shot classification setting is considered, which means that the test set contains new, i.e., "unseen", image classes that had not been included in the training set. Concretely, one image for each of the molecules in a hold-out test set was randomly selected, which means that each unseen molecule class is represented by a single image. Then, samples from this set as well as samples corresponding to both the same molecule and plate were removed from the full test set, in order to ensure that the classification was not due to plate effects. We will refer to the remaining samples as the "test set". Finally, the set of unseen classes consisted of 2115 images in the random split and 1398 in the scaffold split. Overall, the test set comprised 43,778 samples for the random split, and 28,248 for the scaffold split.

To obtain the embeddings, all samples were pushed through the CLOOME image encoder and normalized. Then, cosine similarity was computed between embeddings from the former and the latter, and the softmax function was applied. The output from the softmax function was then used to calculate the metrics shown in Table 3.

We compared the image embeddings of CLOOME to embeddings of a microscopy image encoder trained in a supervised fashion and to CellProfiler features computed as detailed in the "Methods" section. Regarding the image embeddings of GapNet, the images were encoded using the model weights provided by Hofmarcher et al.[20], removing the last layer of its classifier, which resulted in a 1024-dimension embedding space.

As shown in Table 3, CLOOME exhibits higher performance in top-1, top-5 and top-10 accuracy metrics in comparison to both GapNet and CellProfiler-extracted features. Note that the number of classes in the scaffold split is lower than in the random split, such that classifying the correct molecule becomes an easier task, which explains the higher accuracy in the former data split.

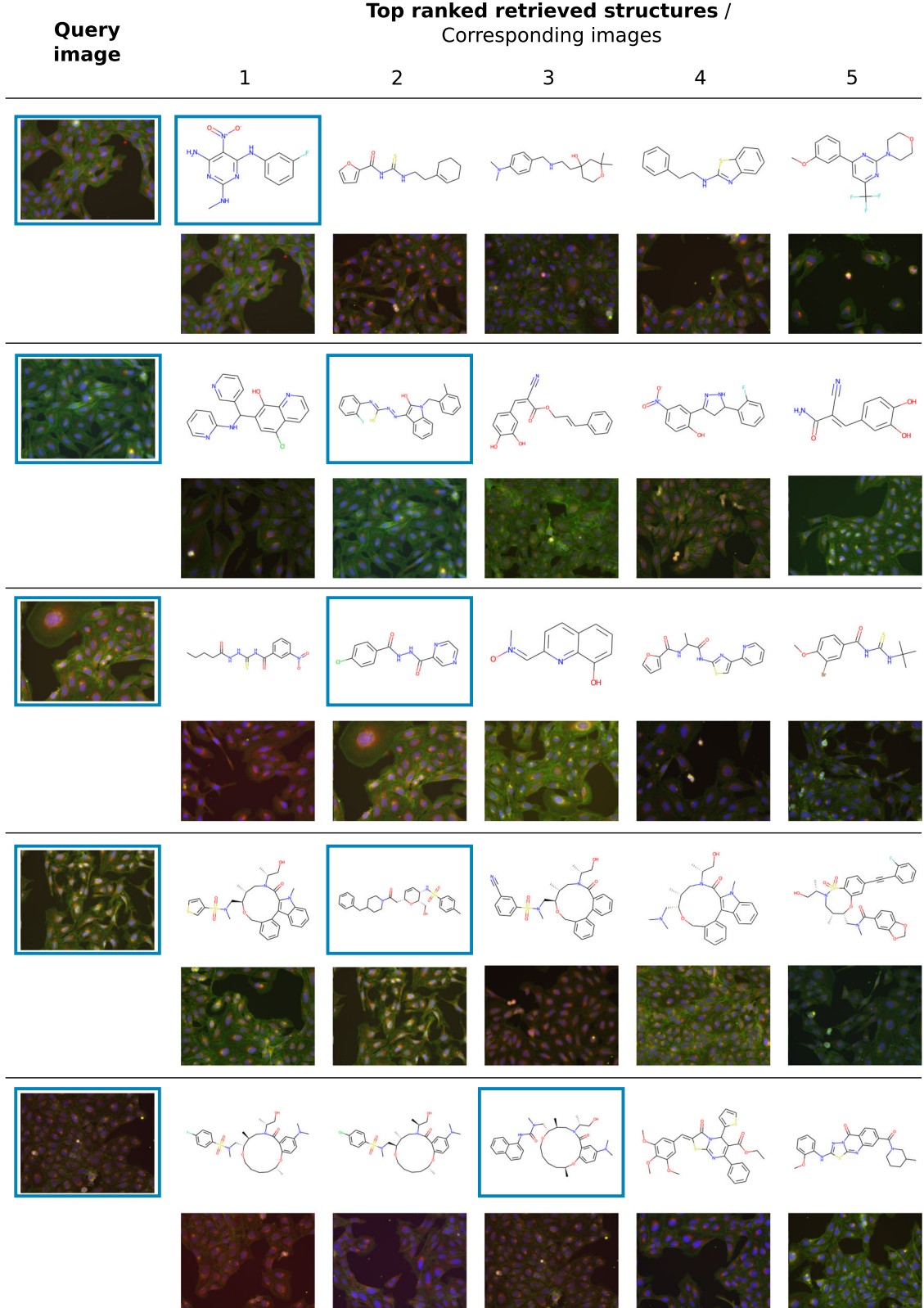

**Fig. 5 | Example results for the retrieval task.** On a hold-out test set, the five molecules for which representations are the most similar to the query image are shown along with their corresponding images. Blue boxes mark the query image and its matched molecular structure, i.e., the matched pair. CLOOME can be used to retrieve molecules that could produce similar biological effects on treated cells, i.e., bioisosteres.

**Table 2 | Comparison of the linear probing evaluation of the learned representations against fully supervised methods[20]**

| Type | Method | AUC | F1 | AUC > 0.9 | AUC > 0.8 | AUC > 0.7 |
|---|---|---|---|---|---|---|
| Linear probing | CLOOME | **0.714** ± 0.20 | 0.395 ± 0.32 | 57 | 84 | 109 |
| | CellProfiler | 0.655 ± 0.20 | 0.273 ± 0.32 | 35 | 63 | 84 |
| Supervised | ResNet | **0.731** ± 0.19 | 0.508 ± 0.30 | 68 | 94 | 119 |
| | DenseNet | 0.730 ± 0.19 | 0.530 ± 0.30 | 61 | 98 | 121 |
| | GapNet | 0.725 ± 0.19 | 0.510 ± 0.29 | 63 | 94 | 117 |
| | MIL-Net | 0.711 ± 0.18 | 0.445 ± 0.32 | 61 | 81 | 105 |
| | M-CNN | 0.705 ± 0.19 | 0.482 ± 0.31 | 57 | 78 | 105 |
| | SC-CNN | 0.705 ± 0.20 | 0.362 ± 0.29 | 61 | 83 | 109 |
| | FNN | 0.675 ± 0.20 | 0.361 ± 0.31 | 55 | 71 | 90 |

For each method the performance metrics area under the receiver operating characteristic curve (AUC) and F1-score are shown, along with their standard deviation ($n = 209$ tasks), and the number of tasks with an AUC higher than 0.9, 0.8, and 0.7. Note that the CLOOME encoders do not have access to any activity data. The features produced by the CLOOME encoder are still predictive for activity data as shown by fitting a logistic regression model, considered as linear probing. CLOOME reaches the performance of the several supervised methods, which indicates transferability of the learned representations[23]. The best method in each category is marked in bold.

**Table 3 | Results for the zero-shot image-to-image molecule classification task**

| Split | Method | Accuracy [%] | | | | | |
|---|---|---|---|---|---|---|---|
| | | Top-1 | 95% CI | Top-5 | 95% CI | Top-10 | 95% CI |
| Random | CLOOME | **18.4** | [18.1, 18.8] | **41.5** | [41.1, 42.0] | **56.2** | [55.8, 56.7] |
| | GapNet | 0.361 | [0.307, 0.422] | 1.07 | [0.973, 1.17] | 1.78 | [1.66, 1.91] |
| | CellProfiler | 2.68 | [2.53, 2.84] | 8.24 | [7.99, 8.51] | 12.6 | [12.3, 12.9] |
| Scaffold | CLOOME | **22.3** | [21.9, 22.9] | **50.8** | [50.2, 51.4] | **67.1** | [66.6, 67.7] |
| | GapNet | 0.517 | [0.437, 0.608] | 1.62 | [1.47, 1.77] | 2.56 | [2.38, 2.75] |
| | CellProfiler | 3.64 | [3.42, 3.86] | 10.4 | [10.0, 10.7] | 15.6 | [15.1, 16.0] |

Given a molecule-perturbed microscopy image, the image corresponding to the matched molecule must be selected from a set of candidates. Top-1, top-5, and top-10 accuracy in percentage are shown for a hold-out test set, along with the upper and lower limits for a 95% confidence interval (CI) ($n = 43,778$ for the random split and $n = 28,248$ for the scaffold split) on the resulting proportion. The best method in each category is marked in bold.

## Zero-shot image-to-image mechanism of action (MoA) classification

We now apply CLOOME to a challenging zero-shot MoA-classification task. This task assesses how well distinguished these embeddings are according to their mechanism of action. In order to evaluate this, labels were taken from the Drug Repurposing Hub[26] of molecules that are present in our validation and test sets.

The same procedure as in the zero-shot molecule classification task (see Subsection above) was followed, with the difference that the set of unseen classes consists of one image per mechanism of action. Again, images corresponding to the same molecule and plate as those from the representative class set were removed in order to ensure that the classification was not influenced by plate effects.

For the random split, the representative set consisted of 126 MoAs and 202 molecules. For the scaffold, this set contained 68 MoAs and 93 molecules. Regarding the test set, it has 8826 samples for the random split and 4056 for the scaffold split. We employed the same baselines as in the zero-shot molecule classification task.

Results displayed in Table 4 show that CLOOME presents better performance than GapNet and CellProfiler features in mechanism of action prediction.

## Statistics and reproducibility

All confidence intervals reported in this study were calculated using the Clopper-Pearson interval. To compute these intervals, the number of test samples in each task were used and have been reported in their specific subsections in "Results".

## Discussion

We have introduced a contrastive learning method for learning representations of microscopy images and chemical structures. On the largest available dataset of this type, we demonstrate that the encoders of CLOOME can be used as a powerful cross-modal retrieval system between chemical structures and bioimages. Additionally, we demonstrated that the CLOOME embeddings are both rich and transferable representations. This opens the possibility to re-use the learned representations for activity or property prediction and for other tasks, such as retrieval tasks from microscopy image or chemical databases, which we demonstrate in a series of challenging downstream tasks.

*Limitations.* Our method currently has several limitations. Our trained networks are restricted to a particular type of microscopy images, which are acquired with the Cell Painting protocol[16]. This protocol has been published and currently there are community efforts[27] to increase the amount of available data. Large and more diverse datasets of molecule-perturbed cells or internal pharmaceutical company datasets will likely improve the learned representations, both image and structure encoder[28], making them more transferable to images from other sources. However, we show in Supplementary Tables 8, 9, 10, 11, and 12 that our method is robust to various distortions (see Supplementary Figure 1) that were not considered during pre-training. Due to the computational complexity, the hyperparameter and architecture space is currently under-explored such that we expect our method to further improve with better hyperparameters or encoder architectures. Furthermore, it has not escaped our notice that the learned structure encoder could also be used for transfer learning on molecular activities and properties. Also, it is worth noting that, although linear probing has been extensively used for the purpose of evaluating the quality of representations[10,11], if the latter are very high dimensional, this method presents the risk of overfitting[22]. While the main focus of this study is a system that retrieves images from a bioimaging database, an alternative line of research on deep generative models exists. Deep generative models focus on generating, rather than retrieving, images or molecules based on an input. Some

**Table 4 | Results for the zero-shot image-to-image mechanism of action (MoA) classification task**

| Split | Method | Accuracy [%] | | | | | |
|---|---|---|---|---|---|---|---|
| | | Top-1 | 95% CI | Top-5 | 95% CI | Top-10 | 95% CI |
| Random | CLOOME | **13.0** | [12.3, 13.7] | **35.3** | [34.3, 36.3] | **48.0** | [46.9, 49.0] |
| | GapNet | 1.09 | [0.882, 1.33] | 4.68 | [4.25, 5.14] | 8.63 | [8.06, 9.24] |
| | CellProfiler | 2.87 | [2.53, 3.24] | 9.46 | [8.86, 10.1] | 14.9 | [14.2, 15.7] |
| Scaffold | CLOOME | **18.5** | [17.3, 19.7] | **47.9** | [46.4, 49.5] | **61.3** | [59.7, 62.8] |
| | GapNet | 1.92 | [1.52, 2.39] | 9.00 | [8.14, 9.92] | 17.2 | [16.1, 18.4] |
| | CellProfiler | 4.54 | [3.92, 5.22] | 15.3 | [14.2, 16.4] | 24.5 | [23.2, 25.8] |

Given a molecule-perturbed microscopy image, the image corresponding to the matched molecule must be selected from a set of candidates. Top-1, top-5, and top-10 accuracy in percentage are shown for a hold-out test set, along with the upper and lower limits for a 95% confidence interval (CI) ($n = 8826$ for the random split and $n = 4056$ for the scaffold split) on the resulting proportion. The best method in each category is marked in bold.

# Image/structure pre-training

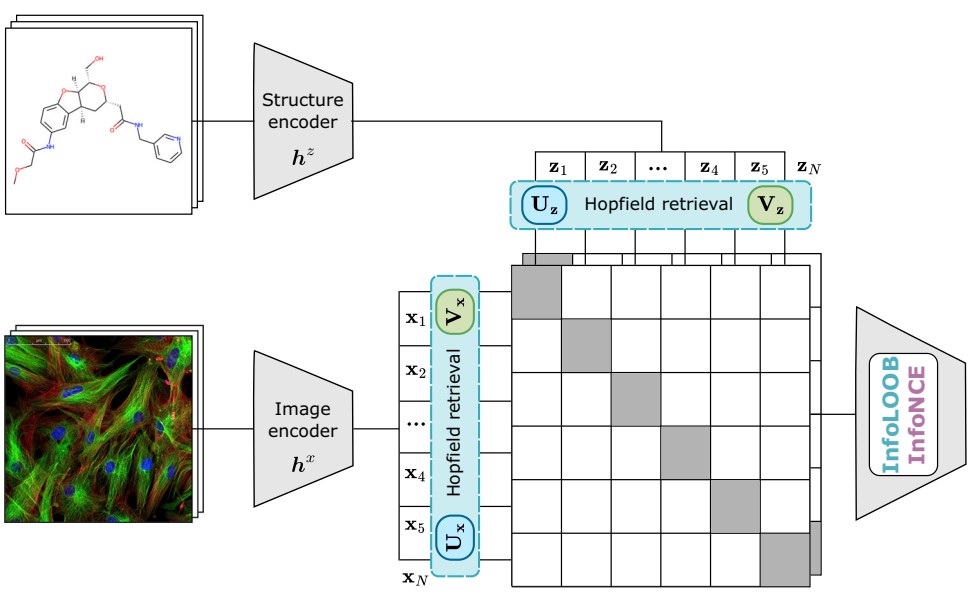

**Fig. 6 | Schematic representation of CLOOME.** Contrastive pre-training of embeddings of the two modalities, microscopy image and chemical structure, of a molecule using the CLOOB[11] or CLIP[10] approach. The training dataset consists of $N$ pairs of microscopy images of molecule-perturbed cells and chemical structures of molecules $\{(x_1, z_1), …, (x_N, z_N)\}$. We assume that an adaptive *image encoder* $\boldsymbol{h}^x(.)$ and an adaptive *structure-encoder* $\boldsymbol{h}^z(.)$ are available that map the microscopy images and chemical structures to their embeddings $\mathbf{x}_n = \boldsymbol{h}^x(x_n)$ and $\mathbf{z}_n = \boldsymbol{h}^z(z_n)$, respectively. In the CLIP approach, these embeddings are directly used to compute the InfoNCE loss. In the CLOOB approach, image and structure embeddings are retrieved from stored image embeddings $\mathbf{U}$ and structure embeddings $\mathbf{V}$, such that $\mathbf{U_x}$ denotes image-retrieved image embeddings, $\mathbf{U_z}$ structure-retrieved image embeddings, $\mathbf{V_x}$ image-retrieved structure embeddings and $\mathbf{V_z}$ structure-retrieved structure embeddings. In this case, the InfoLOOB loss is computed with the latter embeddings.

examples of methods already developed for this purpose are generative adversarial networks conditioned on CellProfiler feature vectors[29] and generative flow methods conditioned on molecules[30]. Having addressed these limitations, we nevertheless believe that both the retrieval system and representations obtained with CLOOME could be highly useful for both the community using bioimaging as well as for drug discovery.

## Methods
### Contrastive Learning and leave-One-Out-boost for Molecule Encoders
We propose contrastive learning of representations from pairs of microscopy images and chemical structures to obtain a common embedding space of these two modalities, a retrieval system and highly transferable encoders (see Fig. 6). In contrast to previous approaches, in which chemical structure encoders learned representations using activity data[19,20] or microscopy image encoders used hand-crafted representations[17,24] or learned on manual annotations,

CLOOME optimizes representations without activity data or human expertise.

The training dataset consists of $N$ pairs of microscopy images of molecule-perturbed cells and chemical structures of molecules $\{(x_1, z_1), …, (x_N, z_N)\}$. We assume that an adaptive *image encoder* $\boldsymbol{h}^x(.)$ and an adaptive *structure-encoder* $\boldsymbol{h}^z(.)$ are available that map the microscopy images and chemical structures to their embeddings $\mathbf{x}_n = \boldsymbol{h}^x(x_n)$ and $\mathbf{z}_n = \boldsymbol{h}^z(z_n)$, respectively. Note that the original image is denoted as $x_n$, which is mapped to an image embedding $\mathbf{x}_n$ by a neural network $\boldsymbol{h}^x(.)$, e.g., a ResNet. The stacked microscopy image embeddings are denoted as $\mathbf{X} = (\mathbf{x}_1, …, \mathbf{x}_N)$ and the stacked structure embeddings as $\mathbf{Z} = (\mathbf{z}_1, …, \mathbf{z}_N)$. The embeddings are normalized such that $\|\mathbf{x}_n\| = \|\mathbf{z}_n\| = 1 \; \forall \, n$ as in other contrastive learning approaches[23]. For notation, see also Supplementary Table 1.

In a contrastive learning setting, methods aim at increasing the similarity of matched pairs and decrease the similarity of un-matched pairs. This task has often been approached by maximizing the mutual information of the embeddings using the InfoNCE loss[9,10,23], which is

also used in the CLIP approach[10]. The InfoNCE objective function has the following form:

$$\mathbf{L}_{\text{InfoNCE}} = -\frac{1}{N}\sum_{i=1}^{N}\ln\frac{\exp(\tau^{-1}\mathbf{x}_i^T\mathbf{z}_i)}{\sum_{j=1}^{N}\exp(\tau^{-1}\mathbf{x}_i^T\mathbf{z}_j)} - \frac{1}{N}\sum_{i=1}^{N}\ln\frac{\exp(\tau^{-1}\mathbf{x}_i^T\mathbf{z}_i)}{\sum_{j=1}^{N}\exp(\tau^{-1}\mathbf{x}_j^T\mathbf{z}_i)}, \quad (1)$$

where $\tau^{-1}$ is the inverse temperature parameter, which scales the cosine similarity of embedding pairs. This parameter can be regarded as a penalization of un-matched samples that are highly similar to the matched one. The lower this parameter is, the more the un-matched, but similar samples influence the loss. Conversely, as this value increases, all pairs will tend to contribute equally to the loss[31].

The contrastive learning method CLIP has the problem of "explaining away"[11,32,33]. Explaining away describes the effect in which few features are over-represented while others are neglected. This effect can be present (a) when learning focuses only on few features and/or (b) when the covariance structure in the data is insufficiently extracted. Explaining away can be caused by saturation of the InfoNCE objective[11,34,35]. To ameliorate these drawbacks, CLOOB[11] has introduced the InfoLOOB objective together with continuous modern Hopfield networks[36] as a promising method for contrastive learning. Our contrastive learning framework CLOOME comprises both methods CLIP[10] and CLOOB[11].

For our extension of the CLOOB method, first image- and structure-embeddings are retrieved from stored image embeddings $\mathbf{U}$ and structure embeddings $\mathbf{V}$. $\mathbf{U}_{\mathbf{x}_i}$ denotes an image-retrieved image embedding, $\mathbf{U}_{\mathbf{z}_i}$ a structure-retrieved image embedding, $\mathbf{V}_{\mathbf{x}_i}$ an image-retrieved structure embedding and $\mathbf{V}_{\mathbf{z}_i}$ a structure-retrieved structure embedding. In analogy to CLOOB, these retrievals from continuous modern Hopfield networks are computed as follows:

$$\mathbf{U}_{\mathbf{x}_i} = \mathbf{U}\,\text{softmax}(\beta\mathbf{U}^T\mathbf{x}_i), \quad (2)$$

$$\mathbf{U}_{\mathbf{z}_i} = \mathbf{U}\,\text{softmax}(\beta\mathbf{U}^T\mathbf{z}_i), \quad (3)$$

$$\mathbf{V}_{\mathbf{x}_i} = \mathbf{V}\,\text{softmax}(\beta\mathbf{V}^T\mathbf{x}_i), \quad (4)$$

$$\mathbf{V}_{\mathbf{z}_i} = \mathbf{V}\,\text{softmax}(\beta\mathbf{V}^T\mathbf{z}_i), \quad (5)$$

where $\beta$ is a scaling parameter of the Hopfield network which is considered a hyperparameter. A $\beta$ value of 0 corresponds to retrieving the average of the stored embeddings. Conversely, when $\beta$ is set to a large value, the stored embeddings that are most similar to the query embeddings $\mathbf{x}_i$ and $\mathbf{z}_i$ are retrieved. The retrieved embeddings $\mathbf{U}_{\mathbf{x}_i}, \mathbf{U}_{\mathbf{z}_i}, \mathbf{V}_{\mathbf{x}_i}, \mathbf{V}_{\mathbf{z}_i}$ are normalized to unit norm. By default, we store the current mini-batch in the continuous modern Hopfield networks, that is, $\mathbf{U} = \mathbf{X}$ and $\mathbf{V} = \mathbf{Z}$. Note that $\mathbf{X}$ contains the image embeddings ($\mathbf{Z}$ the structure embeddings) and we use $N$ ambiguously both as dataset size, but also as mini-batch size to keep the notation uncluttered. The choice that $\mathbf{U} = \mathbf{X}$ and $\mathbf{V} = \mathbf{Z}$ is mostly taken because of computational constraints, while $\mathbf{U}$ and $\mathbf{V}$ could hold the whole dataset or, alternatively, exemplars. Then, the InfoLOOB objective[11,37] for the retrieved embeddings is used as objective function:

$$\mathbf{L}_{\text{InfoLOOB}} = -\frac{1}{N}\sum_{i=1}^{N}\ln\frac{\exp\left(\tau^{-1}\mathbf{U}_{\mathbf{x}_i}^T\mathbf{U}_{\mathbf{z}_i}\right)}{\sum_{j\neq i}^{N}\exp\left(\tau^{-1}\mathbf{U}_{\mathbf{x}_i}^T\mathbf{U}_{\mathbf{z}_j}\right)} - \frac{1}{N}\sum_{i=1}^{N}\ln\frac{\exp\left(\tau^{-1}\mathbf{V}_{\mathbf{x}_i}^T\mathbf{V}_{\mathbf{z}_i}\right)}{\sum_{j\neq i}^{N}\exp\left(\tau^{-1}\mathbf{V}_{\mathbf{x}_j}^T\mathbf{V}_{\mathbf{z}_i}\right)}. \quad (6)$$

**Microscopy image encoder.** Microscopy images differ from natural images in several aspects, for example the variable number of channels that depends on the staining procedure[15,20]. Although standard

image encoders, such as Residual Networks[38] could be in principle used with minor adjustments, alternative approaches, such as multiple instance learning, could be required for very high-resolution datasets[39].

**Molecule structure encoder.** Since the advent of Deep Learning, a large number of architectures to encode molecules have been suggested[40–44]. In contrast to computer vision and natural language processing, in which only few prominent architectures have emerged, there is yet no standard choice for chemical structure encoders. Because of their computational efficiency and good predictive performance, CLOOME uses a descriptor-based fully-connected network[45,46] with 4 hidden layers of 1024 units with ReLU activations and batch normalization (for further details see Supplementary Section 1.2). However, also any graph[43,47–49], message-passing[50], or sequence-based[51] neural network with an appropriate pooling operation can be used as structure encoder.

**Dataset and preprocessing.** *Cell painting*. We use matched pairs of microscopy images and molecules from the Cell Painting[16,24] dataset. This dataset is a collection of high-throughput fluorescence microscopy images of U2OS cells treated with different small molecules[16]. The dataset consists of 919,265 five-channel images corresponding to 30,616 different molecules. The experiment to obtain the microscopy images was conducted using 406 multi-well plates, and each one of the before mentioned individual images are views from a sample spanning the space in the corresponding well, so that six adjacent views belong to one single sample. After disregarding erratic images (out of focus or containing high fluorescence material) as well as images of untreated cells that were used as controls, our final dataset comprises 759,782 microscopy images treated with 30,404 different molecules.

*Preprocessing*. We followed the preprocessing protocol of Hofmarcher et al.[20], which consisted of converting the original TIF images from 16-bit to 8-bit, simultaneously removing the 0.0028% of pixels with highest values.

Moreover, the images were normalized using the mean and standard deviation calculated for the training split. Concerning molecules, two types of preprocessing yielded the best results. For the bioactivity prediction and zero-shot tasks, the SMILES strings were transformed to 1024-bit Morgan fingerprints with a radius of 3, taking chirality into account[52,53]. For the retrieval task, a max-pooling combination of Morgan and RDKit count-based fingerprints, with a final length of 8192 bits, provided better results.

*Data splits*. We split our dataset into training, validation, and test set, using the splits of Hofmarcher et al.[20]. Samples that have not been used in the previous study due to missing activity data, are assigned to the training split. Note that all images belonging to the same molecular structure are moved into the same set. Finally, training, validation and test set consist of 674,357, 28,632, and 56,793 image and molecule pairs, respectively.

**Pre-training, architecture, and hyperparameters.** We use the suggested hyperparameters of OpenCLIP[54] and CLOOB[11] wherever applicable, and tuned a few critical hyperparameters, such as learning rate and the $\beta$ parameter of the Hopfield layer on the validation set. The architecture of the structure encoder was inspired by previous successful models[46] and was not subject to substantial hyperparameter optimization. We used the Adam optimizer[55] with decoupled weight decay regularization[56]. The value for weight decay was 0.1. For the learning rate scheduler, we used cosine annealing with a warm-up of 20,000 steps and hard restarts every 7 epochs[57]. We set the dimension of the embedding space to $d = 512$, which determines the size of the output of both encoders. We use a batch size of 256 as default due to computational constraints.

For the retrieval and zero-shot image classification tasks, a higher validation performance was achieved by a CLIP-like architecture directly using the embeddings returned from the image and structure encoders and the InfoNCE loss. In this case, the inverse temperature parameter $\tau^{-1}$ was set to 14.3 and images were cropped to a pixel resolution of $520 \times 520$, based on performance in the validation set.

For activity prediction as downstream task, the inverse temperature parameter $\tau^{-1} = 30$ was used. For the Hopfield layers, the scaling hyperparameter $\beta = 22$ was selected, and the model was trained for 63 epochs based on linear probing results in the corresponding validation set. For data augmentation and to allow large batch sizes, for the bioactivity prediction task, the images were cropped and re-scaled from the original $520 \times 696$ pixel resolution to $320 \times 320$ during training, whereby the original aspect ratio was mostly maintained.

Hence, different pre-training settings have been found to yield best results for bioactivity prediction and for both the retrieval and zero-shot image classification task, respectively. However, the large majority of hyperparameters were shared in both strategies. Because of the limited exploration of the vast hyperparameter space, we expect potential improvements from further investigations.

For further details on the hyperparameter selection, see Supplementary Tables 2, 3, 4 and 5.

**CellProfiler features calculation and preprocessing.** The CellProfiler software[17] can also be considered as a microsopy image encoder that supplies image embeddings. These features were calculated using the latest CellProfiler pipelines provided in the CellPainting gallery[58]. The resulting embeddings consist of 1240 features aggregated in one vector per image. Features were aggregated to allow comparability across all methods' results. After filtering out features with a standard deviation of zero across all samples as well as trivial features (e.g., file paths), these embeddings consist of 1081 features. Before training, these features were standardized using the mean and standard deviation calculated for the embeddings corresponding to the training set.

### Related work

**Contrastive learning has had a strong impact on computer vision and natural language processing.** Over the last decade, supervised deep learning methods have achieved important advances in the field of computer vision[38,59]. These supervised methods require large amounts of labeled data, which may be very costly or unfeasible to obtain, and they have limited generalization abilities[60,61]. This has led to the exploration of new methods that are able to learn robust representations of the data which can be transferred to different downstream tasks[23,62]. With contrastive learning methods[63] and self-supervision these meaningful representations can be obtained without the need for large amounts of expensive manually-provided labels[23,64–66]. While uni-modal methods typically use pre-text tasks[23], for multi-modal methods the self-supervision arises from the availability of two modalities of an instance, such as image and text[10,67]. Both uni-modal and multi-modal contrastive learning methods have recently had a substantial impact in computer vision and natural language processing[68].

**CLIP for multi-modal data yields remarkable performance at zero-shot transfer learning and has recently been improved by CLOOB.** A well-established multi-modal approach is Contrastive Language-Image Pre-training (CLIP)[10], which learns both image- and text-representations simultaneously. CLIP shows comparable performance to methods that are solely image-based and yields highly transferable representations, which is shown by its high performance at zero-shot transfer learning. However, CLIP has recently been shown to suffer from the "explaining away" effect[11,32,33] (details in "Methods"). Considering this caveat, the "Contrastive Leave One Out Boost" (CLOOB) method has been proposed[11]. CLOOB uses a different objective, the "InfoLOOB" (LOOB for "Leave One Out Bound")

objective[37], which does not include the positive pair in the denominator to avoid saturation effects[11]. Moreover, continuous modern Hopfield networks[36] are used to reinforce the covariance structure of the data. As a result, CLOOB has further improved zero-shot transfer learning. The ability to learn transferable representation from multimodal data makes CLOOB the prime candidate for learning representations of molecules in drug discovery.

**Contrastive learning for molecule representations in drug discovery.** In drug discovery, the effect of the limited availability of data on molecules is even more severe, since the acquisition of a single bioactivity data point can cost several thousand dollars and take several weeks or months[69,70]. Therefore, methods that can learn transferable representations from unlabeled data are highly demanded. Thus, several contrastive learning approaches have been recently developed for different tasks in drug discovery. MolCLR[71] uses contrastive molecule-to-molecule training by augmenting molecular graphs. Stärk et al.[72] contrastively learn 3D and 2D molecule representations to inform the learned molecule encoder with 3D information. Lee et al.[73] and Seidl et al.[74] use contrastive learning for molecules and chemical reactions, and Vall et al.[75] utilize text representations of wet-lab procedures to enable zero-shot predictions. However, none of these methods have exploited the wealth of information contained in microscopy images of molecule-perturbed cells[16] and demonstrated strong transferability of the learned molecule encoders.

**Image-based profiling of small molecules has strongly improved the drug discovery process.** Characterizing a small molecule by the phenotypic changes it induces to a cell, is considered promising for accelerating drug discovery[16,19,76,77]. The advantages of this biotechnology are that it is time- and cost-effective as compared to standard activity measurements. Measuring the effects of a molecule on a biological system early in the drug discovery process might be useful to improve clinical success rates[78]. Particularly, microscopy image-based profiles of small molecules have been suggested to be effective together with deep learning methods[77]. However, the current efforts are still in standard supervised learning settings based on extracted features[19] or deep architectures[20]. The amount of labeled images is in the range of few tens of thousands, although international efforts are currently building datasets which are magnitudes larger[27]. Instead of the currently used activity measurements as labels[19,20], we propose a self-supervised contrastive learning strategy of image- and structure-based molecule encoders: Contrastive Leave One Out boost for Molecule Encoders (CLOOME). CLOOME extends recent successful contrastive learning methods to the fields of biological imaging and drug discovery. Our approach intends to overcome the limited transferability of current molecule encoders[79,80].

### Reporting summary

Further information on research design is available in the Nature Portfolio Reporting Summary linked to this article.

## Data availability

The model weights are publicly available in the HuggingFace repository: https://huggingface.co/anasanchezf/cloome/tree/main. The dataset for training is available at: http://gigadb.org/dataset/100351[24]. A script with the functions used to preprocess data as in this study can be found at: https://github.com/ml-jku/cloome/blob/main/src/preprocess/preprocess_image.py. Source data are provided with this paper.

## Code availability

Code is available at: https://github.com/ml-jku/cloome (https://doi.org/10.5281/zenodo.8344964[81]).

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

## Acknowledgements

ASF's research position was funded by the European Union's Horizon 2020 research and innovation program under the Marie Skłodowska-Curie Innovative Training Network - European Industrial Doctorate grant agreement No. 956832, "Advanced machine learning for Innovative Drug Discovery". We thank Markus Hofmarcher for his support on the linear probing using CellProfiler features on the bioactivity prediction task. The ELLIS Unit Linz, the LIT AI Lab, the Institute for Machine Learning, are supported by the Federal State Upper Austria. IARAI is supported by Here Technologies. We thank the projects AI-MOTION (LIT-2018-6-YOU-212), DeepFlood (LIT-2019-8-YOU-213), Medical Cognitive Computing Center (MC3), INCONTROL-RL (FFG-881064), PRIMAL (FFG-873979), S3AI (FFG-872172), DL for GranularFlow (FFG-871302), EPI-LEPSIA (FFG-892171), AIRI FG 9-N (FWF-36284, FWF-36235), ELISE (H2020-ICT-2019-3 ID: 951847), Stars4Waters (HORIZON-CL6-2021-CLIMATE-01-01). We thank Audi.JKU Deep Learning Center, TGW LOGISTICS GROUP GMBH, Silicon Austria Labs (SAL), FILL GmbH, Anyline GmbH, Google, ZF Friedrichshafen AG, Robert Bosch GmbH, UCB Biopharma SRL, Merck Healthcare KGaA, Verbund AG, GLS (Univ.

Waterloo) Software Competence Center Hagenberg GmbH, TÜV Austria, Frauscher Sensonic and the NVIDIA Corporation.

## Author contributions

A.S.F. and E.R. implemented the methods and algorithms. A.S.F. and E.R. designed and performed the experiments. A.S.F., E.R., S.H., and G.K. analyzed and interpreted the results. A.S.F., E.R., and G.K. wrote the manuscript. S.H. and G.K. conceived and designed the study.

## Competing interests

The authors declare no competing interests.
