## [Peer Review File · Nature Communications]

Reviewers' Comments:

Reviewer #1:

Remarks to the Author:

Authors have addressed the comments from previous reviewers very well. The current version is clearly written. The methodology is justified, and evaluation is comprehensive. However, several major issues remain.

1. To rigorously evaluate the practical value of GLOOME, the performance in all tested cases should be evaluated using corrupted images (e.g., rotation) that are not used in the training.
2. Strictly speaking, the "zero-shot" learning evaluation in the manuscript does not follow the conventional definition of "zero-shot learning". As author mentioned, the task is to evaluate if the similar embeddings corresponds to similar chemicals. "zero-shot learning" is better to be re-worded to avoid confusion.
3. Table 4, it is surprising that the random split outperforms scaffold split. An explanation could be helpful.

Reviewer #3:

Remarks to the Author:

In the revised version of Nature Communications, the authors have effectively addressed the majority of my concerns. Notably, they have incorporated additional baselines in the experiments, demonstrating that their method (CLOOME) outperforms these baselines across multiple retrieval tasks. Furthermore, they have provided clearer explanations regarding the practical applications of their work in the Introduction section. Additionally, the web interface has been enhanced to include more information.

My only remaining concern is that the application of this work is limited to the "retrieval" setting. In other words, the proposed method cannot generate new images or novel molecules based on an input molecule or image. In scenarios where data sparsity is an issue, the existing images or molecules in the (small) database may not be sufficient for a given query (molecule/image). I understand that transitioning from "retrieval" to "generation" may not be a straightforward task and may require significant updates to the current method. However, it is important to acknowledge this limitation of CLOOME for drug discovery applications. I would like to refer the authors to existing works that have attempted to generate new images/molecules for the Cell Painting dataset [1, 2].

[1] Yang, Karren, et al. "Improved conditional flow models for molecule to image synthesis." CVPR (2021).

[2] Zapata, Paula A. Marin, et al. "Cell morphology-guided de novo hit design by conditioning GANs on phenotypic image features." Digital Discovery (2023).

Reviewer #4:

Remarks to the Author:

The authors have addressed our comments and concerns from the previous revision in a satisfactory manner. They have added more baselines to the comparisons, extended explanations, and made the document much clearer in terms of methodology, findings, and supporting evidence.

I have one suggestion regarding Table 3. It is imprecise to bold CLOOME accuracy as the best

method, when the "Linear probing" category has a single entry. I suggest adding the results of a linear classifier using the precomputed CellProfiler features. This is expected to perform worse than FNN, but it is a reasonable entry to include in the "Linear probing" category. In this way, it will be accurate to state the CLOOME representation is indeed the best performing.

Other than this, I have no further comments or suggestions.

Responses to the Reviewers

Manuscript NCOMMS-23-05908-T “CLOOME: contrastive learning unlocks bioimaging databases for queries with chemical structures”

Thanks to the Reviewers

We thank the reviewers for their positive, thoughtful and constructive feedback. Their suggestions were included in this new version of the manuscript, and we believe that they helped to further improve it.

Overview of changes

- We extended the Appendix with an evaluation of CLOOME for all downstream tasks using corrupted images, showing CLOOME’s robustness to distortions that were not applied while pre-training.
- We included a new baseline for the bioactivity prediction experiment performing linear probing on CellProfiler features.
- We acknowledged the limitation of CLOOME regarding the generation of molecules and images, and referred to previous studies in this direction.
- We provided further interpretation of the results.

Changes have been highlighted in the PDF file of the revised manuscript.

Responses to Reviewer 1

1. To rigorously evaluate the practical value of GLOOME, the performance in all tested cases should be evaluated using corrupted images (e.g., rotation) that are not used in the training.

Response: We thank the reviewer for their suggestion. We are now reporting the performance of CLOOME when evaluating on corrupted images with different types of distortions that were not applied to the images in the pre-training phase, such as random flipping and rotation.

We report the results for all of the downstream tasks presented in the previous version of the manuscript. Overall, different types of corruptions only slightly change the performance metrics (see Supplementary Section D). Concretely, for the cross-modal retrieval tasks, the performance on corrupted image remains mostly within the confidence intervals of the previous evaluation with the original images, and for the bioactivity prediction the mean AUC changes only from 0.714 to 0.713. For the zero-shot tasks, especially the molecule classification task, the performance is slightly more affected by the applied distortions. A possible explanation is that image embeddings corresponding

to cells treated with different molecules are closer to each other than different structure embeddings. If this is the case, corrupting the images will have a larger impact in image-to-image (i.e. zero-shot) tasks than in cross-modal tasks (i.e. retrieval). Overall, evaluating the model on corrupted images leads to almost the same metrics as before. Analogously to supervised learning, slight drops of the metrics could be mitigated by applying corruptions during the pre-training phase which will make the model more robust to them.

2. Strictly speaking, the “zero-shot” learning evaluation in the manuscript does not follow the conventional definition of “zero-shot learning”. As author mentioned, the task is to evaluate if the similar embeddings corresponds to similar chemicals. “zero-shot learning” is better to be re-worded to avoid confusion.

Response: We thank the reviewer for bringing up this issue of how the term “zero-shot” learning is used in our work and other works. Indeed, “zero-shot” learning evaluation is often done in different ways. Zero-shot evaluation has been defined in literature as measuring the capabilities of a given model to predict classes that have not been seen during training (Xian et al., 2017; Larochelle et al., 2008). As CLOOME was not trained to correctly assign images to their corresponding molecule or MoA, we consider our test cases as zero-shot. However, we also understand that we did not use the same setting as in CLIP, in which they performed image-to-text classification. For clarity, we now specify “image-to-image” for our zero-shot tasks in the manuscript.

3. Table 4, it is surprising that the random split outperforms scaffold split. An explanation could be helpful.

Response: We believe that the reviewer might be referring to the fact that the top-k accuracies of the scaffold split are higher than for the random split, which could seem counterintuitive. As we mention in the paper, in the scaffold split, we do not include molecules that contained the same scaffolds present in either the training set nor validation sets. This results in a lower number of molecules, i.e. classes in this case, in the scaffold split, such that classifying the correct molecule becomes an easier task.

Responses to Reviewer 3

My only remaining concern is that the application of this work is limited to the “retrieval” setting. In other words, the proposed method cannot generate new images or novel molecules based on an input molecule or image. In scenarios where data sparsity is an issue, the existing images or molecules in the (small) database may not be sufficient for a given query (molecule/image). I understand that transitioning from “retrieval” to “generation” may not be a straightforward task and may require significant updates to the current method. However, it is important to acknowledge this limitation of CLOOME for drug discovery applications. I would like to refer the authors to existing works that have attempted to generate new images/molecules for the Cell Painting dataset [1, 2].

[1] Yang, Karren, et al. “Improved conditional flow models for molecule to image synthesis.” CVPR (2021).

[2] Zapata, Paula A. Marin, et al. "Cell morphology-guided de novo hit design by conditioning GANs on phenotypic image features." *Digital Discovery* (2023).

Response: We agree with the reviewer that generative models using microscopy images are indeed an interesting line of research. Although image or molecule generation is out of the scope of this study, we acknowledge the relevance of this kind of methods. We already cited generative models for natural images (Ramesh et al., 2022) based on CLIP models and, in the new version of the paper, we also refer to previous generative methods for high-content screening data.

Responses to Reviewer 4

I have one suggestion regarding Table 3. It is imprecise to bold CLOOME accuracy as the best method, when the "Linear probing" category has a single entry. I suggest adding the results of a linear classifier using the pre-computed CellProfiler features. This is expected to perform worse than FNN, but it is a reasonable entry to include in the "Linear probing" category. In this way, it will be accurate to state the CLOOME representation is indeed the best performing.

Response: We now extended the linear probing experiments as suggested, using CellProfiler features. This category now has more than a single entry and we thank the reviewer for this useful suggestion.

Minor changes

We noticed that the reported values for the zero-shot MoA classification and scaffold split with CLOOME were based on a different split and we updated these values to be comparable to the other methods (marked red in Table 5). The said split contained samples treated with the same molecule and from the same plate as some of the classes. We also corrected two typos and one rounding error. The conclusions that can be drawn from these results do not change with respect to the previous version. We also added an acknowledgment to Markus Hofmarcher in the respective section.

References

- Larochelle, H., Erhan, D., and Bengio, Y. (2008). Zero-data learning of new tasks. In *Association for the Advancement of Artificial Intelligence Conference*.
- Ramesh, A., Dhariwal, P., Nichol, A., Chu, C., and Chen, M. (2022). Hierarchical text-conditional image generation with clip latents.
- Xian, Y., Lampert, C. H., Schiele, B., and Akata, Z. (2017). Zero-shot learning – a comprehensive evaluation of the good, the bad and the ugly. In *Asian Conference on Computer Vision*.

Reviewers' Comments:

Reviewer #1:

Remarks to the Author:

The revision has addressed the concerns. Thanks.

Reviewer #3:

Remarks to the Author:

The authors have sufficiently addressed my comments and improved the paper.

Reviewer #4:

Remarks to the Author:

The authors have satisfactorily addressed my comments. I therefore favorably recommend the manuscript for publication.

Responses to the Reviewers

Manuscript NCOMMS-23-05908-T “CLOOME: contrastive learning unlocks bioimaging databases for queries with chemical structures”

Thanks to the Reviewers

We thank the reviewers for considering and accepting our updates to the feedback they previously gave our manuscript. We believe that the rounds of review helped to improve the paper in great measure.

Responses to Reviewer 1

The revision has addressed the concerns. Thanks.

Response: Thank you for the constructive feedback which helped us a lot to improve our manuscript.

Responses to Reviewer 3

The authors have sufficiently addressed my comments and improved the paper.

Response: Thank you for the constructive feedback which helped us a lot to improve our manuscript.

Responses to Reviewer 4

The authors have satisfactorily addressed my comments. I therefore favorably recommend the manuscript for publication.

Response: Thank you for the constructive feedback which helped us a lot to improve our manuscript.